# Adulteration of beeswax: A first nationwide survey from Belgium

**Noëmie El Agrebi**[1☯], **Lidija Svečnjak**[2☯]*, **Jelena Horvatinec**[2], **Véronique Renault**[1],
**Agnes Rortais**[3], **Jean-Pierre Cravedi**[4], **Claude Saegerman**[1]*

**1** Research Unit for Epidemiology and Risk Analysis applied to veterinary sciences (UREAR-ULiège),
Fundamental and Applied Research for Animals and Health (FARAH) Center, University of Liège, Liege,
Belgium, **2** Faculty of Agriculture, Department of Fisheries, Apiculture, Wildlife Management and Special
Zoology, University of Zagreb, Zagreb, Croatia, **3** Scientific Committee and Emerging Risks Unit, European
Food Safety Authority (EFSA), Parma, Italy, **4** UMR1331 Toxalim (Research Centre in Food Toxicology)
INRAE, ENVT, INP-Purpan, UPS, Toulouse, France

☯ These authors contributed equally to this work.
* claude.saegerman@uliege.be (CS); lsvecnjak@agr.hr (LS)

doi.org/10.1371/journal.pone.0252806

BRAZIL

**Data Availability Statement:** All relevant data are
within the paper and its Supporting Information
files.

**Funding:** Funder #1: Belgian Federal Public Service
of Health, Food Chain Safety, and Environment;

## Abstract

Beeswax is intended for use in the beekeeping sector but also in the agro-food, pharmaceutical or cosmetics sectors. The adulteration of beeswax is an emerging issue that was
reported lately at several occasions in the scientific literature. This issue tends to become
more frequent and global, but its exact extent is not accurately defined. The present study
aims to assess the current situation in Belgium through a nationwide survey. Randomized
beeswax samples originating from Belgian beekeepers (N = 98) and commercial suppliers
(N = 9) were analysed with a Fourier transform infrared spectroscopy (FTIR) coupled with
Attenuated Total Reflectance (ATR) accessory (FTIR-ATR spectroscopy) for adulteration.
The survey revealed a frequency of 9.2% and 33.3% of adulteration in beekeepers beeswax
samples (9 samples out of 98: 2 with paraffin and 7 with stearin/stearic acid) and commercial
beeswax samples (3 samples out of 9: all adulterated with stearin/stearic acid), respectively.
The analysed samples were adulterated with various percentages of paraffin (12 to 78.8%)
and stearin/stearic acid (1.2 to 20.8%). This survey indicates that in the beekeepers samples, beeswax adulteration was more frequent in comb foundation and crude beeswax than
in comb wax. With the example of this nationwide survey conducted in Belgium, this study
shows the emergence of the issue and the urgent need for action to safeguard the health of
both honey bees health and humans, in particular with the setting of a proper regulation
legal framework and a specific routine analytical testing of commercial beeswax to ensure
beeswax quality.

## Introduction

Honey bees (*Apis mellifera* L.) are the main pollinators in agricultural ecosystems [1]. Beeswax
is essential for the beekeeping sector (production of comb foundations) but also for agro-food,
pharmaceutical and cosmetics sectors. In Europe, beeswax is considered as an animal by-

Grant: RF 15/6300 Bee Best Check; Funder #2: SPW ARNE (Service public de Wallonie, Agriculture, Ressources naturelles et Environnement); Grant: RWD32-0286 Bee Tox; Check Funder #3: University of Liège; Grant: FSR-F-VT-16/16 Bee Tox Check; Funder #4: University of Zagreb Grant. The funders had no role in study design, data collection and analysis, decision to publish, or preparation of the manuscript.

**Competing interests:** The authors have declared that no competing interests exist.

product Category 3 material and, therefore, it is not intended for human consumption [2]. However, beeswax is an authorized food additive in the European Union [3] and is a food substance considered as safe according to the U.S. Food & Drug Administration, FDA (21CFR184, 1973) [4]. In some cases, honey is sold with honeycombs to demonstrate its authenticity [5,6], resulting in a dietary exposure to beeswax for consumers eating both the honey and the comb and those exposed to honey comb debris present in honey. As reported by Hargrove et al. [7] consumption of beeswax may reach a few grams per day and per person in a small portion of the population [7], but such dietary exposure could be increased with this practice being more frequently advertised and promoted via internet (online sales of ready-to-eat honeycombs).Therefore, there is a concern that the adulterated beeswax might enter into the food chain (*e.g.* through the use of honeycombs) and present a risk to human health [8].

On a global scale, between 2016 and 2018, a yearly average production of 1.9 million tonnes of honey and 69,000 tonnes of beeswax were registered in the FAOSTAT database [9]. Indeed, managed honey bees colonies represent an important source of goods and income [10]. Despite a slow increase of managed honey bee colonies to face agricultural demand for pollination [11], several monitoring programs indicate a global decline in bee populations around the world (e.g. [12–14]). Multiple stress factors, or drivers [15] affecting honey bees, alone or in combination [16–20] are referred to as a possible explanation of this decline.

Besides, in the recent years, beeswax adulteration with paraffin and/or stearin (e.g. [21–26]) has become a growing and alarming concern. The practice of adulteration is emphasised by the fact that beeswax is often salvaged, re-melted, and reused within the beekeeping sector [27].

However, few representative (randomized survey) and published reports are available on the prevalence, the type and the level of adulteration of beeswax. At European level, the most recent study [28] using an advanced method of detection of adulteration (Fourier transform infrared spectroscopy (FTIR) coupled with Attenuated Total Reflectance (ATR) accessory, so-called FTIR-ATR spectroscopy) revealed that, among 137 samples of comb foundation or wax blocks originating from 15 different countries sampled between 2016 and 2018, 59.9% were adulterated by paraffin and/or stearin. Within these samples, levels of adulteration were comprised between 5–93.5% (for paraffin) and/or stearin (solely in Belgium and The Netherlands representing 7.3% of the samples with a level of adulteration between 18.75 and 31.25%). No trace of other adulterants (e.g. tallow, carnauba wax) were detected.

The effect of beeswax adulteration on honey bee health (especially on brood) and human health (through the consumption of bee products) are currently poorly studied [27,28]. However, in Belgium, adverse effects of adulterated beeswax foundations on bee brood development were recently identified [29–31]. This study showed that adulteration levels as low as 5% and 7.5% of stearic and palmitic acids, respectively led to brood mortality rates above 45%.

According to the EU Food Fraud Network (https://ec.europa.eu/food/safety/food-fraud/ffn_en), a dedicated Network for cross-border non-compliances related to food and feed, adulteration of beeswax, that is intended for honey production, with paraffin and/or stearin is considered as a fraud, when meeting four criteria (violation of Law, intention, economic gain, and consumer deception) [32].

To clarify the situation, the European Commission requested the European Food Safety Authority (EFSA) to define purity criteria for beeswax and to assess the health risks for honey bees and humans [8,31,33].

The present study aims to assess the current situation of beeswax authenticity in Belgium through a nationwide cross-sectional survey. Randomized beeswax samples originating from Belgian beekeepers, and commercial suppliers were analysed for adulteration using FTIR-ATR spectroscopy.

## Materials and methods

### Sample selection

In Belgium, 200 beekeepers were randomly selected from the Federal Agency for the Safety of the Food Chain (FASFC) beekeepers database, which included 4,949 registered beekeepers in 2015. One apiary per beekeeper was sampled for beeswax between May and November 2016. The number of beekeepers was stratified by province. Out of the selected beekeepers (N = 200), 91.5% of them provided a beeswax sample for analysis (N = 182).

To be able to detect an expected minimum prevalence of 3% of adulteration with a confidence level of 95%, and considering a population size of 4,949 registered beekeepers, we estimated the sample size for this survey at 98 beekeepers. Indeed, a sub-sample of 98 samples was randomized, and further submitted to the laboratory for analysis of adulterants (i.e. paraffin, and stearin/stearic acid). All sampled bee colonies seemed healthy, with no clinical signs of infectious diseases or acute intoxication.

Eight beeswax (comb foundation) samples randomly collected from different commercial suppliers in Belgium and one additional, achieved by the FASFC, from a Chinese batch of beeswax (2015) where mosaic brood was reported, were analysed for adulteration. All samples were kept in hermetic plastic bags and stored at -20°C until analysis.

### Sample preparation

Comb wax samples collected from the beekeepers were melted by boiling water prior to further analysis in order to remove hive-originating impurities and homogenize the samples into crude beeswax. In case of significant contamination (e.g. significant amount of residues of cocoons in brood combs), samples were re-melted 2–3 times until they were completely purified. Crude beeswax samples and comb foundations were analysed as obtained.

### Beeswax adulteration detection by FTIR-ATR spectroscopy

Preparation of in-house reference material (genuine/authentic beeswax, adulterants, and adulterant-beeswax mixtures containing different proportions of adulterants) for calibration purposes, was performed according to the procedure described in a chapter of the BEEBOOK manual on standard methods for *A. mellifera* beeswax research by Svečnjak et al. [34] (see section "6.2.5.1. Generating IR spectral database of reference samples") with a modification of preparing the adulterant-beeswax mixtures by following 5% increasing sequence of adulterant addition (instead of originally proposed 10%) to improve precision in detecting adulterants in beeswax. For this, in total 38 adulterant-beeswax mixtures were prepared: 18 paraffin-beeswax mixtures (containing 5 to 95% of paraffin; *Paraffinum solidum*, Ph.Eur. 7,8, Kemig, Croatia), and 18 stearic acid-beeswax mixtures (containing 5 to 95% of stearic acid; *Acidum stearicum*, Ph.Eur. 8.1, Kemig, Croatia). Mixtures were placed in a temperature chamber for 3h at 90°C for melting and homogenization. Pure paraffin, pure stearic acid, as well as genuine (pure) beeswax, were subjected to the same temperature treatment in the same way as adulterant-beeswax mixtures.

Beeswax samples were analysed by Fourier transform infrared spectroscopy (FT-IR) using an Attenuated Total Reflectance (ATR) recording technique. Infrared (IR) spectra of investigated beeswax samples were acquired using Cary 660 Fourier transform mid-infrared spectrometer (Agilent Technologies, Palo Alto, CA, USA) with a DTGS (deuterated triglycine sulphate) detector and CsI (cesium iodide) optics, coupled with Golden Gate high temperature (up to 200°C) heated single-reflection diamond ATR accessory (Specac).

FTIR-ATR spectra of prepared in house reference material and collected Belgian beeswax samples were recorded under the same conditions (in the liquid state at 75°C; spectral range: 4000–400 cm$^{-1}$; spectral resolution: 4 cm$^{-1}$; 64 scans/spectrum) in accordance with the method described by Svečnjak et al. [34] in the BEEBOOK section "5.3.2. Analysis of beeswax by IR spectroscopy/5.3.2.1. FTIR-ATR recording technique".

Raw spectral data were stored and pre-analyzed using the software package Resolutions Pro version 5.3.0 (2015) (Agilent Technologies, Palo Alto, CA, USA). Further chemometric modelling and statistical analyses were performed using the software package specialized for spectral data analysis—Origin version 8.1 (Origin Lab Corporation, Northampton, MA, USA). Prediction strength and prediction error of Calibration model were estimated by the simple linear regression whereas prediction strength and prediction error in detecting the adulteration level were determined, i.e. coefficient of determination ($R^2$) and standard error (SE). Quantification of adulterants in beeswax was carried out automatically using the instrument software (Resolutions Pro) after establishing and evaluating the calibration procedure.

## Epidemiological analysis

**Data on bee mortality.** The sampling in beekeepers (N = 98) was conducted jointly with a questionnaire to record colony losses and management practices. The total loss rate (winter and seasonal) was calculated by dividing the total number of colonies lost between September 2015 and April 2016 by the number of colonies in September 2015 multiplied by 100 [35] excluding removed, sold, and purchased colonies.

**Mapping.** The map (**Fig 1**) was produced by a co-author (VR) with quantum-GIS. The GPS data for the country and regional boundaries originate from a copyright free website: DIVA-GIS | free, simple & effective (diva-gis.org). The coordinates of the sample points were collected during the survey and registered into an Excel file. They have been projected with quantum GIS on the country layer and the map. This is therefore an original map with no copyright issues.

**Statistical analyses.** The percentage and corresponding 95% confidence interval (95% CI) of paraffin and stearin/stearic acid adulteration was estimated using an exact binomial distribution [36].

Two logistic regressions were performed. The first one was done using both samples form beekeepers and commercial suppliers (N = 107) and for which the information on the type of beeswax was available (i.e. comb wax as a reference group, comb foundation and crude beeswax from beekeepers and another beeswax from the commercial suppliers). The second one was done using only samples from beekeepers (N = 98) for which more information was available. For the second one, a univariate logistic regression model was used to explain adulteration expressed as binary dependent variable ("1" as adulterated and "0" as non-adulterated beeswax samples). The following exploratory variables were considered: the type of beeswax (categorical variable, which includes comb wax as a reference group, comb foundation, and crude beeswax), the year of introduction of the beeswax in the hive (categorical variable), the province of origin of the beekeepers (categorical variable), and the colony loss rate (continuous variable).

For the type of beeswax, the following definition was used: (i) comb wax (beeswax from old combs from the brood chamber provided by some beekeepers), (ii) comb foundation (beeswax foundation present in beekeepers as a mixture of beeswax from different trade origins), (iii) crude beeswax (melted old brood, and/or honey wax combs, and or cappings to be reused), and (iv) beeswax form suppliers (foundation sold by suppliers).

For the colony loss rate, two binary levels were considered: "0" for colony mortality rates ≤10%, and "1" for colony mortality rates >10% [37].

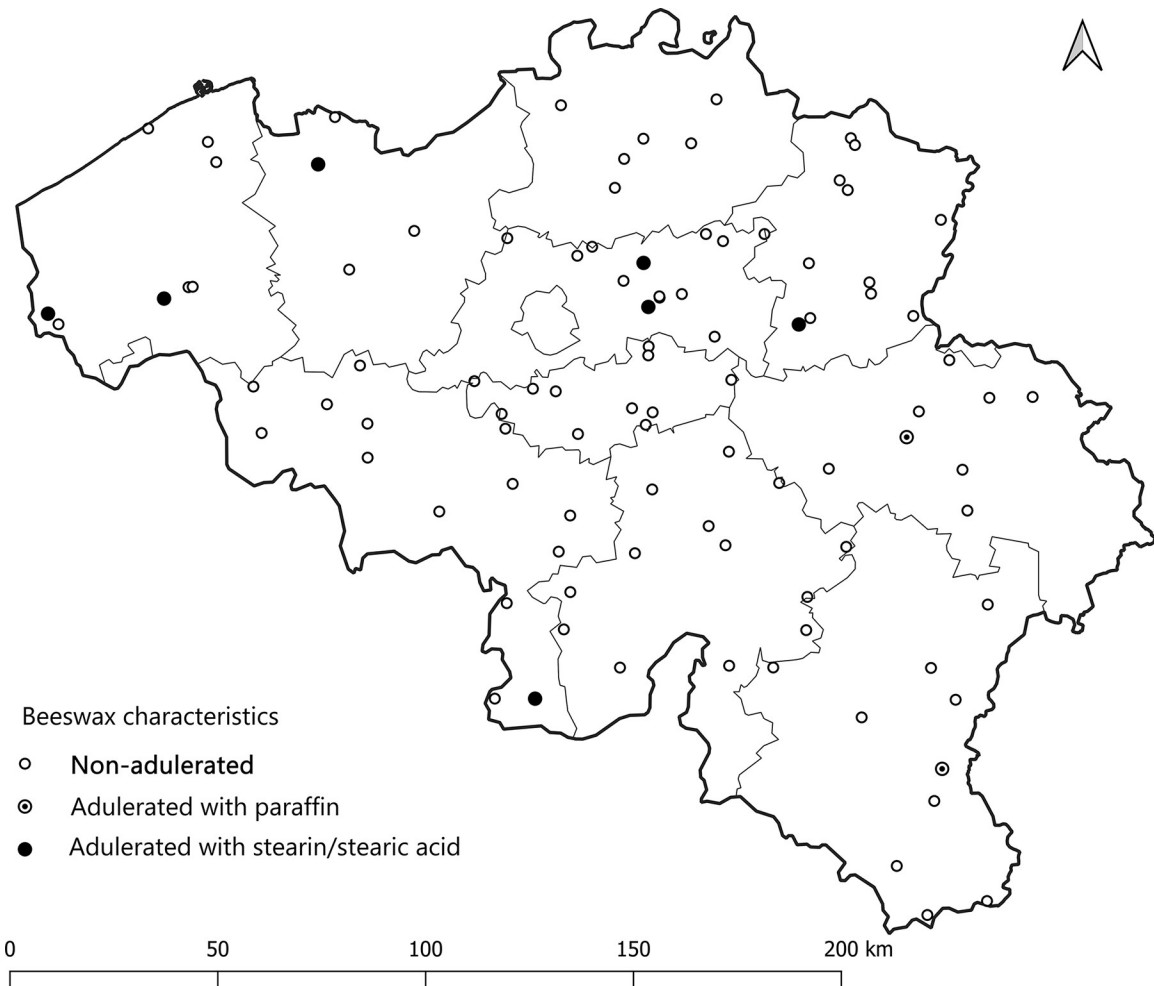

Beeswax characteristics

○ Non-adulerated

◉ Adulerated with paraffin

● Adulerated with stearin/stearic acid

**Fig 1. Location of the samples provided by the beekeepers (N = 98).** The thin line represents the subdivision of the Belgian provinces.

Then, a multivariate logistic regression was performed using the most significant variables ($p$-value < 0.2) out of the univariate model. The use of the Firth logit method allowed inference of odds ratios and 95% confidence interval (CI) when complete separation (zero-cells) occurred [38]. Finally, in a backward stepwise multivariate model, the least significant variable (with the highest $p$-value) were eliminated in a step-by-step approach. At each stage, a likelihood ratio test was used to compare the complex and simplified models. When there was no significant difference between them (using value of $P > 0.05$), the simplified model was used. The goodness of fit of the final multivariate model was assessed using the Hosmer-Lemeshow goodness-of-fit test [36]. All models and tests were performed using Stata SE 14.1® (StataCorp LP, College Station, TX, USA), and the limit of statistical significance of performed tests was defined as 0.05.

## Results

### Indirect validity of the randomization of beeswax samples provided by some beekeepers

For this cross-sectional survey, the sample size used was of 98 samples out of the 182 original beeswax samples (see materials and methods). For this reason, the representativeness of the

sample subset was tested at the province level, related to the whole sample dataset after randomization. The 98 samples of beeswax represent accurately the whole sample dataset (Fisher's exact test (df = 9); $p$-value = 0.69).

## Adulteration of randomized beeswax samples provided by some beekeepers

The samples (N = 98) were randomized, and collected from each Belgian province to be analysed for adulteration (Fig 1). The level of adulteration in analysed beeswax samples was determined based on the IR spectra of reference standards (genuine beeswax, adulterants, and adulterant-beeswax mixtures containing different proportions of adulterants) and calibration curves generated for prepared adulterant-beeswax mixtures. As presented in Fig 2, FTIR-ATR spectra of beeswax and different types of adulterants (an example of paraffin and stearic acid) exhibit specific spectral features with the most prominent and indicative absorption bands in the fingerprint region (1800-800 cm$^{-1}$). IR spectra of prepared adulterant-beeswax mixtures containing 5 to 95% (w/w) of adulterants, i.e. paraffin-beeswax mixtures (Fig 3), and stearic acid-beeswax mixtures (Fig 4), also revealed a specific trend of spectral alterations reflected in decreasing (following the addition of paraffin) and increasing (following the addition of stearic acid) intensities of absorption bands related to esters and free fatty acids. Two spectral regions with target peak areas showing the best correlation between the instrument response and known proportions of adulterant in the adulterant-beeswax reference standards were chosen for further calibration process and quantification of adulterants in analysed beeswax samples.

For paraffin, a target peak area 1750–1727 cm$^{-1}$ (with an absorption maximum at 1738 cm$^{-1}$) and 1198–1147 cm$^{-1}$ (with an absorption maximum at 1171 cm$^{-1}$) showed the best prediction performance (Pearson's r = 0.9994, R$^2$ = 0.9987, SE = 0.00097—**Figs 5 and 6**, and Pearson's r = 0.9996, R$^2$ = 0.9993, SE = 0.00017—**Figs 7 and 8**, respectively), and were therefore used for detecting the paraffin share in analysed beeswax samples. The amount of stearic acid in analysed beeswax samples was estimated based on 1721–1707 cm$^{-1}$ (with an absorption maximum at 1710 cm$^{-1}$) and 1308–1253 cm$^{-1}$ target peak areas (with an absorption maximum at 1281 cm$^{-1}$) that revealed the best prediction performance parameters, i.e. Pearson's r = 0.9994, R$^2$ = 0.9987, SE = 0.00111—**Figs 9 and 10**, and Pearson's r = 0.9999, R$^2$ = 0.9999, SE = 0.00005—**Figs 11 and 12**, respectively. The amount of adulterants (as %, w/w) in analysed beeswax samples was determined as an average value of instrument response for the above-mentioned reference peaks for each adulterant type, i.e. paraffin and stearic acid. Given that stearic acid and a widespread cheap substance called "stearin" (commercially available as a mixture of stearic and palmitic acid, or even as a pure stearic acid) exhibit almost the same spectral features (S1 Fig), the same calibration curve can be used for the detection of both substances. Therefore, the terminology stearin/stearic acid is used further in the text.

Beeswax samples were adulterated with paraffin (N = 2) and stearin/stearic acid (N = 7), but no multi-adulteration was observed. Also, no traces of other adulterants (such as tallow and carnauba wax) or other foreign substances were detected (S2A Fig). Indeed, the level of adulteration of beeswax samples provided by some beekeepers was calculated as 2.04% (95% confidence interval [CI]: 0.25–7.18), and 7.14% (95% CI: 2.92–14.16%) for paraffin and stearin/stearic acid, respectively. The level of beeswax adulteration with paraffin was 12% and 78.8% (Fig 13). The level of beeswax adulteration with stearin/stearic acid (N = 7; i.e. 1.2, 2.2, 2.3, 2.4, 7, 8.1 and 11.9%, respectively) (Fig 14).

## Adulteration of beeswax samples from commercial suppliers (trade wax)

The eight wax samples collected from different commercial suppliers, and the one additional, achieved by the FASFC, were analysed for adulteration. None of the tested samples was

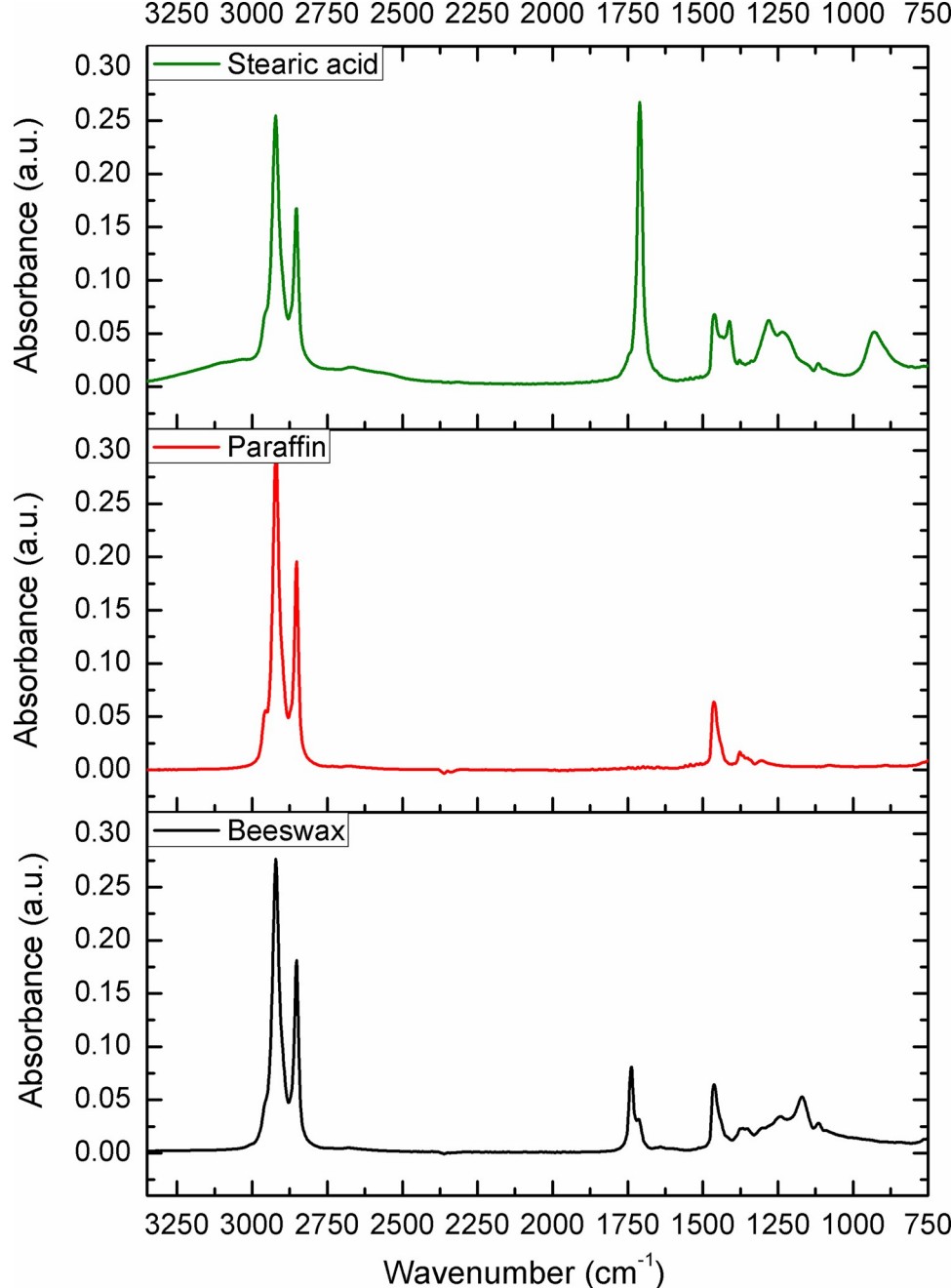

**Fig 2. FTIR-ATR spectra of reference standards used for calibration—genuine (pure) beeswax, and adulterants (paraffin—*Paraffinum solidum*, stearic acid—*Acidum stearicum*).** Wavenumber, the number of waves per unit distance; cm, centimetre; a.u. is for the absorbance unit.

adulterated with paraffin but 3 of them (33%) were adulterated with stearin/stearic acid. The adulteration percentages were 1.5, 3, and 20.8%, respectively (**Fig 15**). The most adulterated sample containing a level of 20.8% of stearin/stearic acid corresponds to the one where mosaic brood was reported. The IR spectra of other comb foundations analysed (N = 6) revealed no trace of other adulterants (**S2B Fig**).

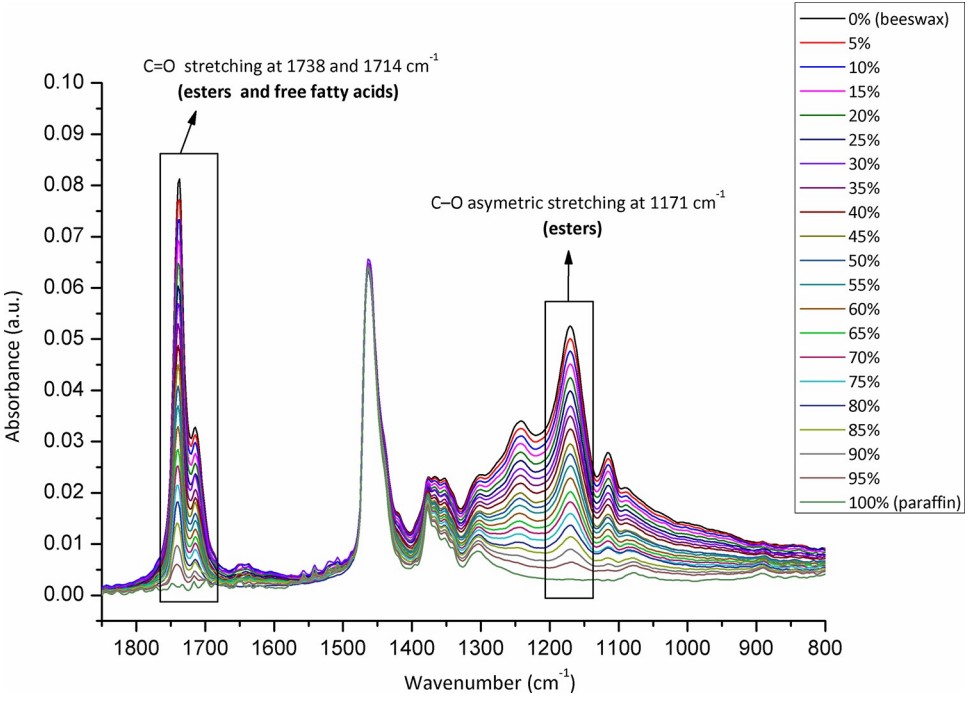

**Fig 3. FTIR-ATR spectra of reference standards (paraffin-beeswax mixtures containing different proportions of paraffin) used for calibration.** Wavenumber, the number of waves per unit distance; cm, centimetre; a.u. is for the absorbance unit.

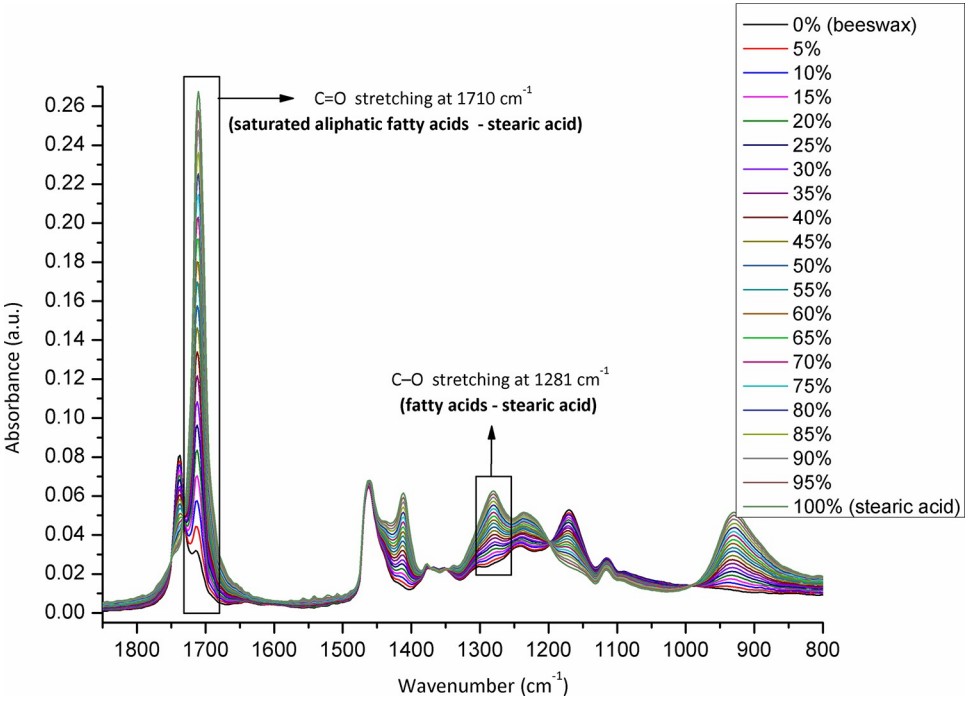

**Fig 4. FTIR-ATR spectra of reference standards (stearic acid-beeswax mixtures containing different proportions of stearic acid) used for calibration.** Wavenumber, the number of waves per unit distance; cm, centimetre; a.u. is for the absorbance unit.

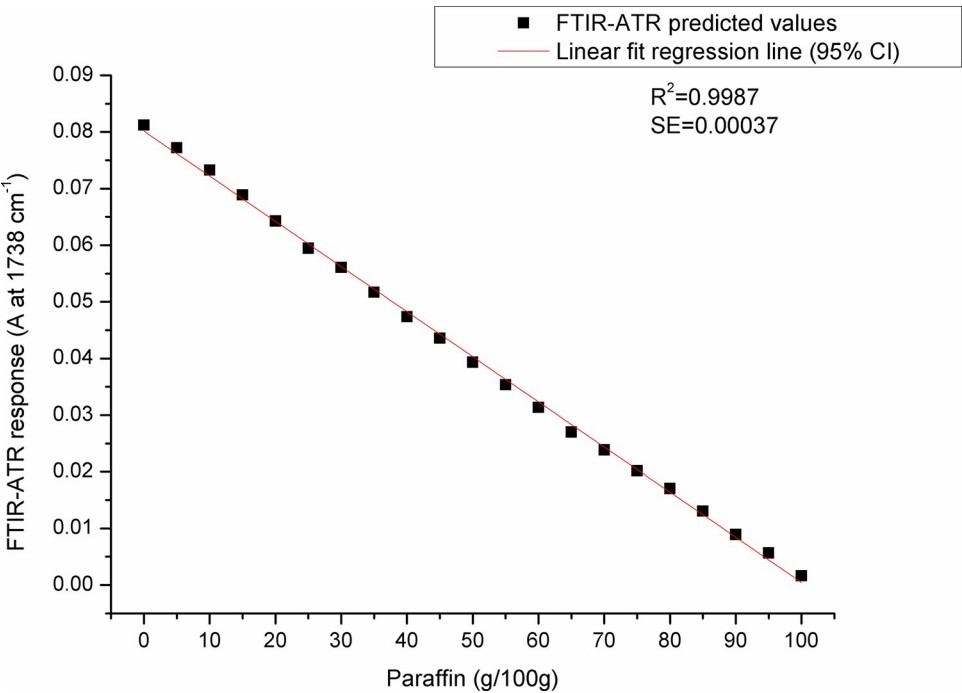

**Fig 5. Prediction performance parameters of the calibration curve constructed for determination of the paraffin share in beeswax: A scatter plot of FTIR-ATR predicted values (instrument response) versus real (known) paraffin share values using the spectral region with an absorption maximum at 1738 cm⁻¹.**

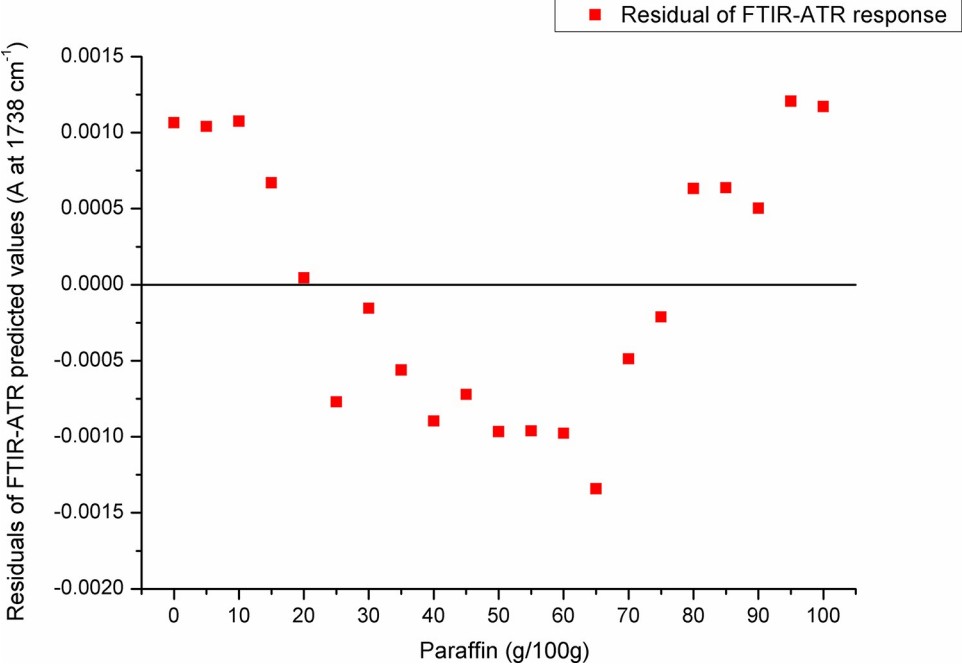

**Fig 6. Residuals of FTIR-ATR prediction in the spectral region with an absorption maximum at 1738 cm⁻¹.**

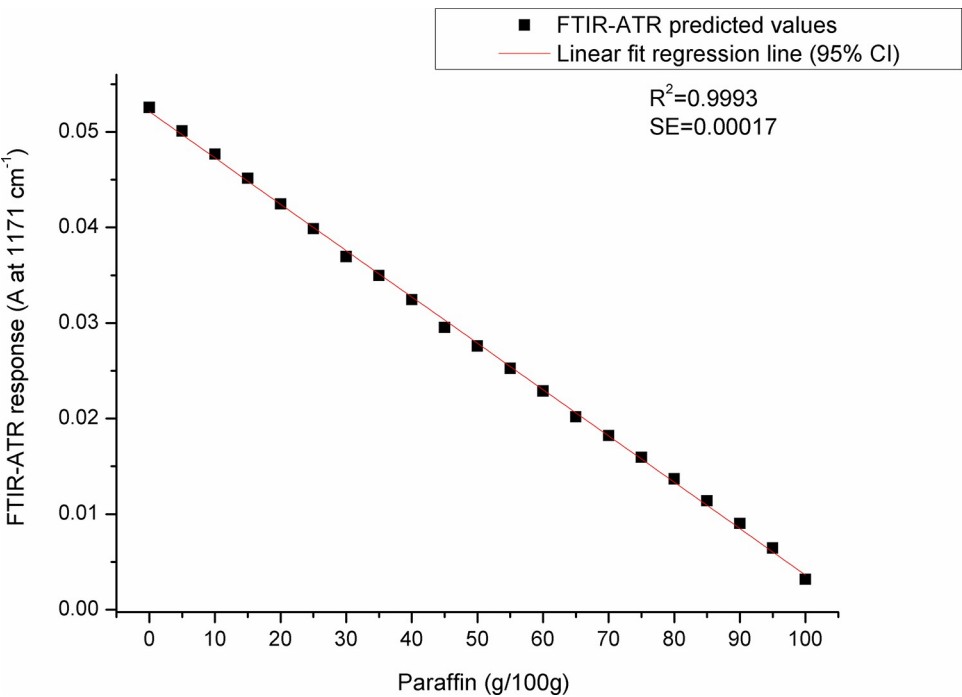

**Fig 7. A scatter plot of FTIR-ATR predicted values (instrument response) versus real (known) paraffin share values using the spectral region with an absorption maximum at 1171 cm$^{-1}$.**

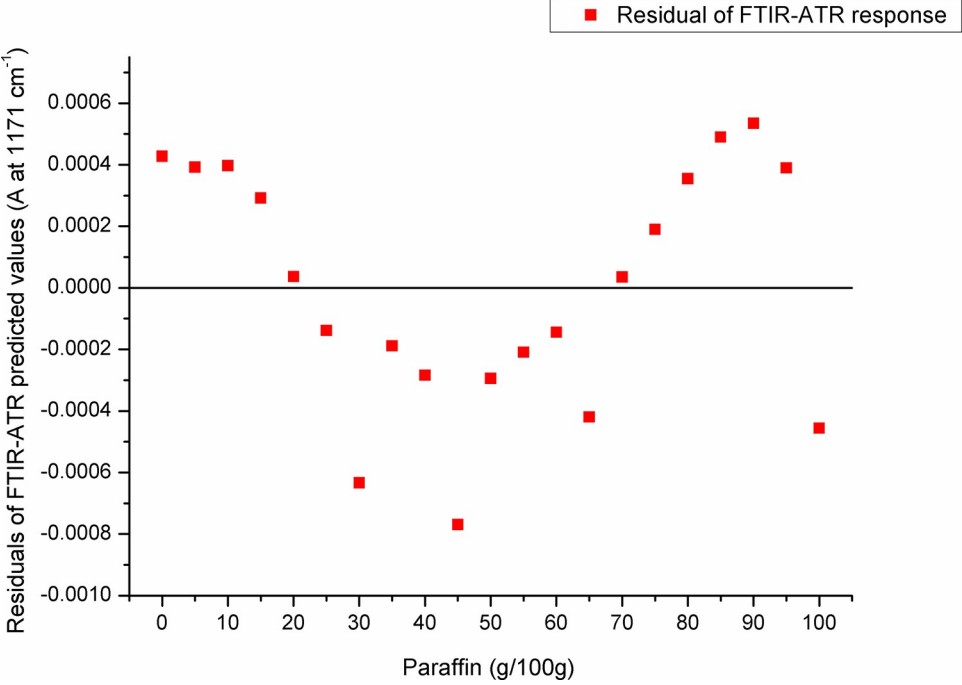

**Fig 8. Residuals of FTIR-ATR prediction in the spectral region with an absorption maximum at 1171 cm$^{-1}$.**

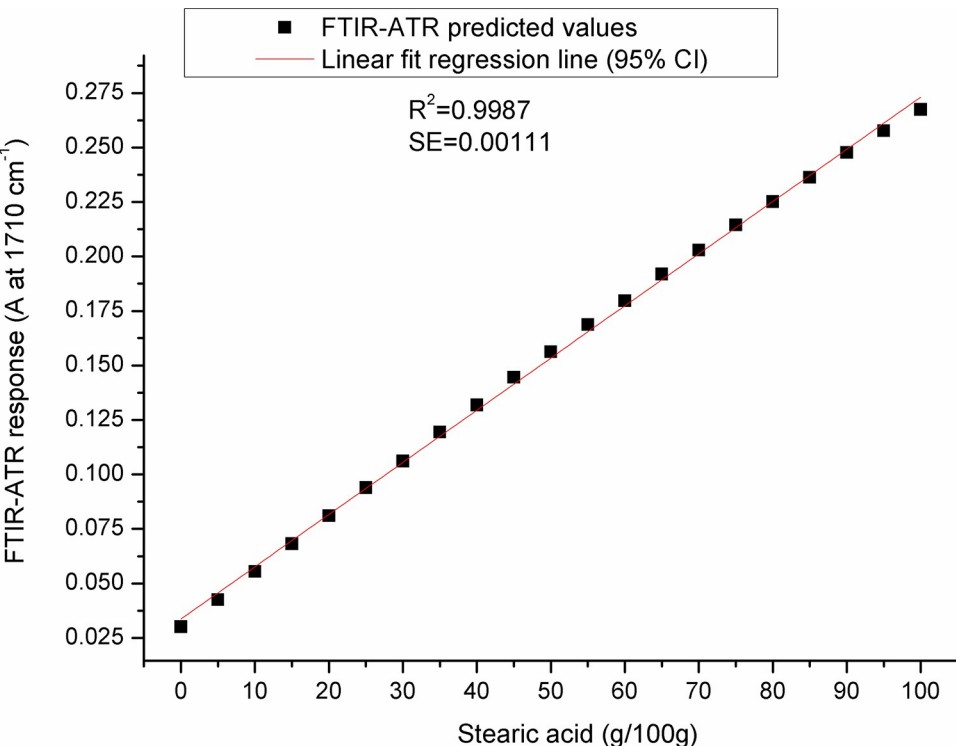

**Fig 9. Prediction performance parameters of the calibration curve constructed for determination of the stearic acid share in beeswax: A scatter plot of FTIR-ATR predicted values (instrument response) *versus* real (known) stearic acid share values using the spectral region with an absorption maximum at 1710 cm$^{-1}$.**

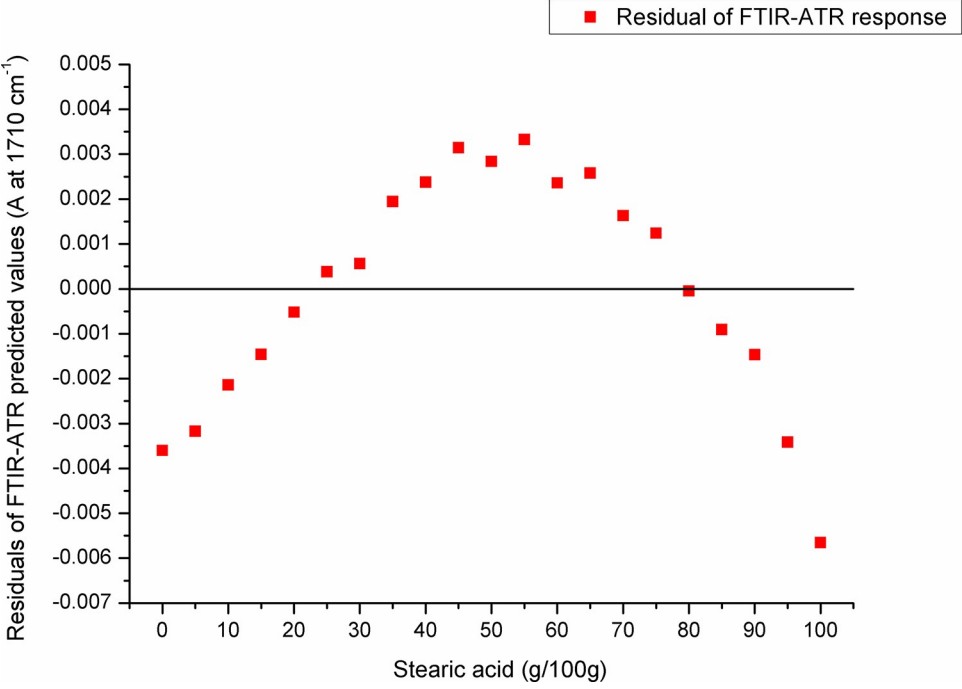

**Fig 10. Residuals of FTIR-ATR prediction in the spectral region with an absorption maximum at 1710 cm$^{-1}$.**

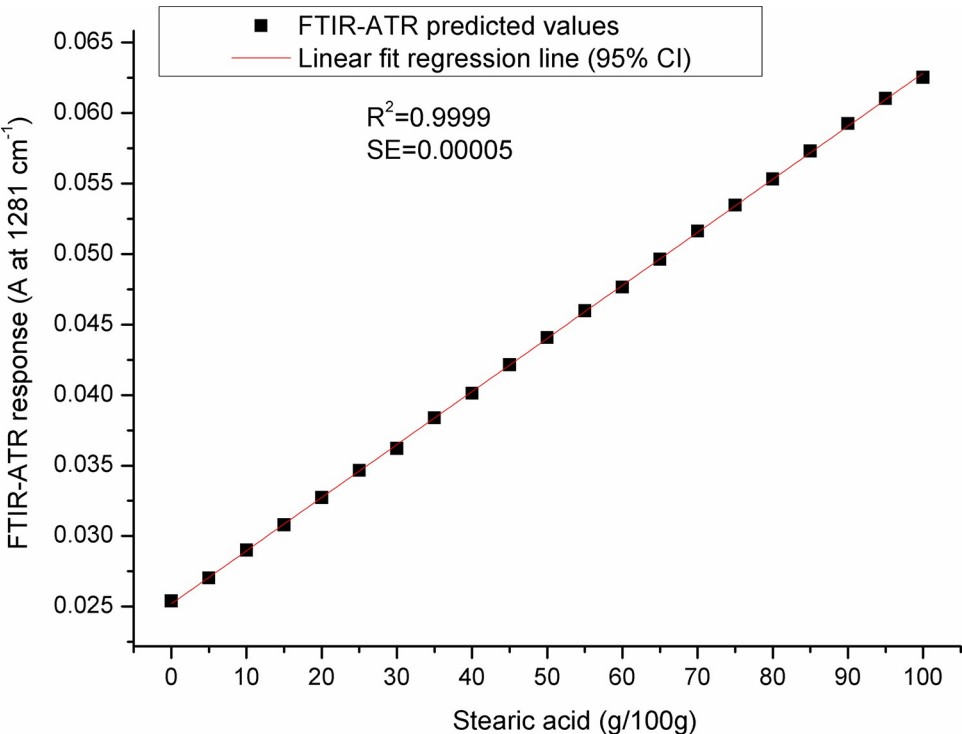

**Fig 11. A scatter plot of FTIR-ATR predicted values (instrument response) versus real (known) stearic acid share values using the spectral region with an absorption maximum at 11281 cm$^{-1}$.**

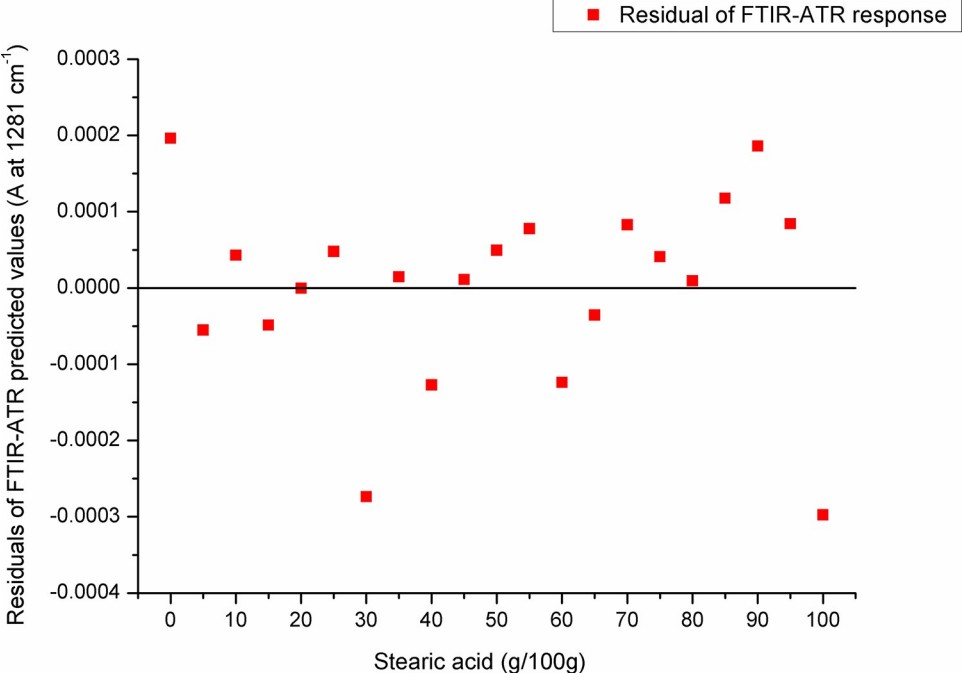

**Fig 12. Residuals of FTIR-ATR prediction in the spectral region with an absorption maximum at 1281 cm$^{-1}$.**

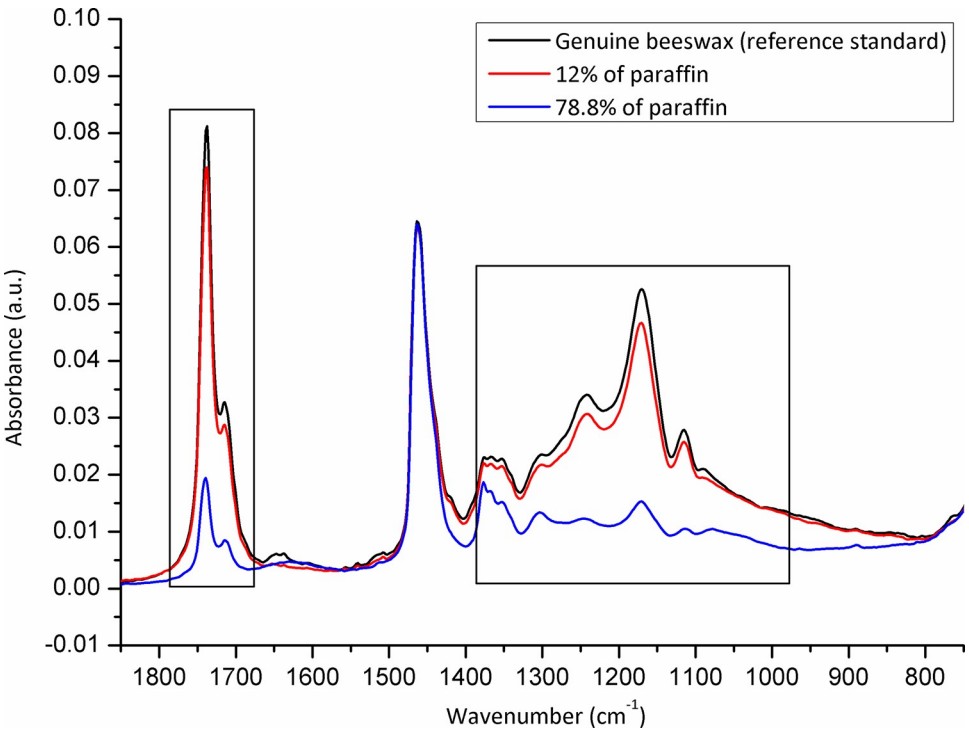

**Fig 13. Adulterated beeswax samples *versus* genuine beeswax (reference standard) with an emphasis on spectral regions indicative for adulteration detection: Paraffin-adulterated beeswax samples.** Due to some spectra overlaps (close share of spectra to 2%), only spectra with more than 2% of difference were presented in this figure. Wavenumber, the number of waves per unit distance; cm, centimetre; a.u. is for the absorbance unit.

### Logistic regression analysis

Due to a low percentage of paraffin and stearin/stearic acid adulteration found in the beeswax samples coming from the beekeepers, the two types of adulterants were considered in the same logistic regression analysis.

**Beeswax samples from both beekeepers and commercial suppliers.** Adulteration is more likely to occur in crude beeswax (OR = 7.70; 95% CI: 1.45–40.93; *p*-value = 0.017) and in comb foundation (OR = 14.75; 95% CI: 2.04–106.46: *p*-value = 0.008) than in comb wax as a reference group (**Table 1**).

**Beeswax samples form beekeepers.** In both of the univariate (**Table 2**) and the multivariate analyses, only one exploratory variable was related to adulterated beeswax samples, i.e. the type of beeswax. Indeed, adulteration is more likely to occur in crude beeswax (OR = 7.70; 95% CI: 1.45–40.93; *p*-value = 0.017) compared to comb wax as a reference group (**Table 2**). The Hosmer–Lemeshow test showed that the final model fits the data well (Chi$^2$ = 0.00, df = 1, *p*-value = 1).

### Discussion

The presence of adulteration of beeswax by paraffin or stearin/stearic acid from samples collected in Belgium was confirmed using a randomized cross-sectional nationwide survey. Based on a logistic regression analysis, using both paraffin and stearin/stearic acid (due to the relatively limited number of positive samples), significantly more adulteration was found in crude beeswax and comb foundation samples than in comb wax as a reference group.

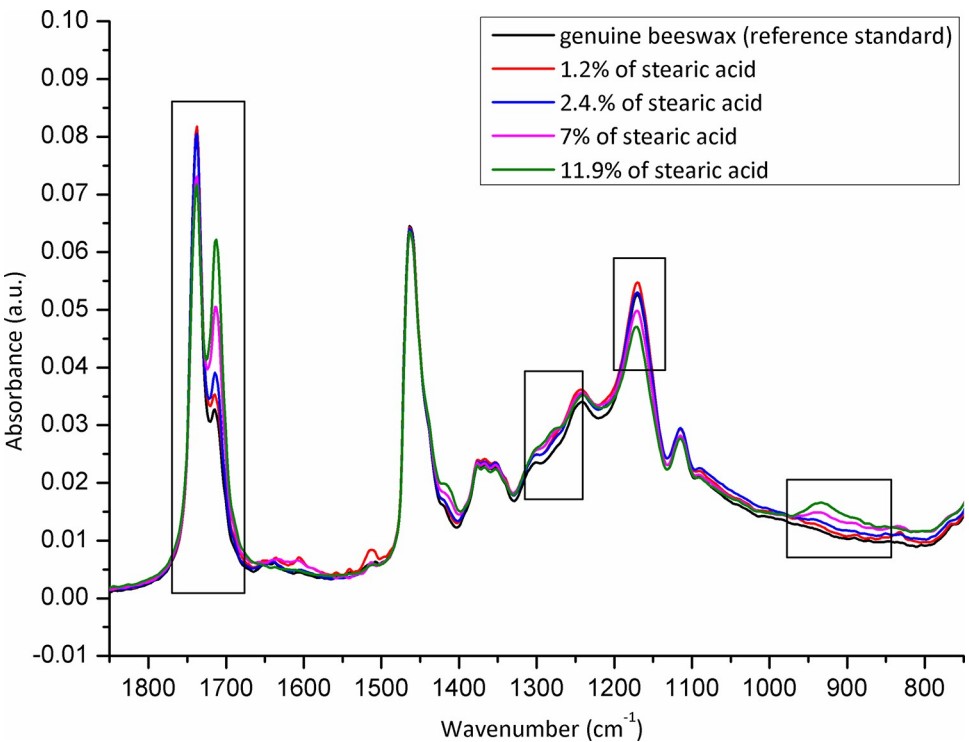

**Fig 14. Adulterated beeswax samples *versus* genuine beeswax (reference standard) with an emphasis on spectral regions indicative for adulteration detection: Stearic acid—adulterated beeswax samples.** Due to some spectra overlaps (close share of spectra to 2%), only spectra with more than 2% of difference were presented in this figure. Wavenumber, the number of waves per unit distance; cm, centimetre; a.u. is for the absorbance unit.

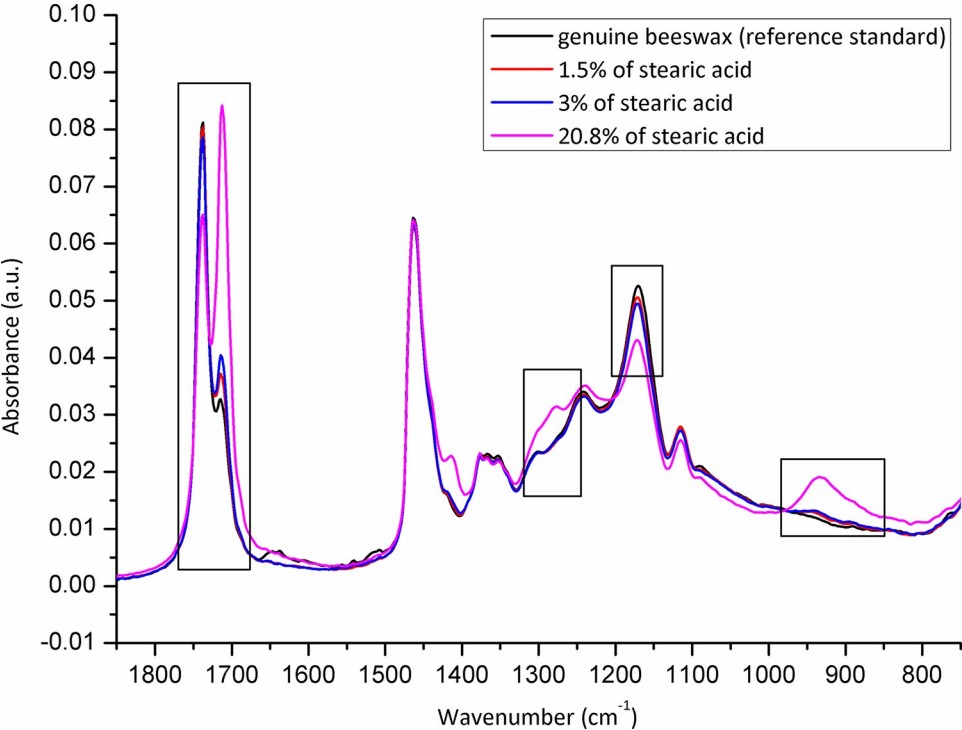

**Fig 15. Comb foundation samples adulterated with stearic acid (N = 3) *versus* genuine beeswax (reference standard) with an emphasis on spectral regions indicative for adulteration detection.** Wavenumber, the number of waves per unit distance; cm, centimetre; a.u. is for the absorbance unit.

**Table 1. Contingency table of results for adulteration of beeswax.**

| Origin of samples | Type of beeswax | Adulterated | Non-adulterated | Total |
|---|---|---|---|---|
| Beekeepers | Comb wax | 2 | 59 | 61 |
| | Comb foundation | 1 | 7 | 8 |
| | Crude beeswax | 6 | 23 | 29 |
| Commercial suppliers | Comb foundation | 3 | 6 | 9 |
| | Total | 12 | 95 | 107 |

This result demonstrates that beekeepers should preferentially use and recycle their own waxes (e.g. cappings wax) rather than using trade wax, following good management practices for wax recycling. In addition, it shows the need for more appropriate guidelines for beeswax production, trade and sale. Beeswax traceability and authentication should be conducted with regular surveillance beekeeping programs. To conduct such surveillance programs, the determination and use of purity criteria (using physico-chemical methods) for beeswax intended for use in beekeeping should be implemented [33]. The use of more advanced methods (e.g. FTIR-ATR spectroscopy) should be promoted, and risk-based survey (e.g. based on trade business of beeswax, and/or by identification, and tracking of emerging risks from beeswax adulteration in the media as recently suggested by Rortais et al. [8]) should be designed and performed.

Despite the use of an advanced analytical method (i.e. FTIR-ATR spectroscopy) with a limit of detection <3% [25,39], the percentage of adulterated beeswax samples coming from beekeepers is relatively low (9.2% for both paraffin, and stearin/stearic acid) in comparison with commercial beeswax samples (33.3% in this survey). Even with a limited number of analysed commercial beeswax samples, the adulteration percentage is similar to the percentage detected

**Table 2. Univariate logistic regression analysis for adulterated *versus* non-adulterated Belgian beeswax samples.**

| Variable | Modalities | Odds ratio | (95% CI) | *p*-value |
|---|---|---|---|---|
| Beeswax type | Comb wax | Reference | - | - |
| | Comb foundation | 4.21 | (0.34–52.64) | 0.264 |
| | Crude beeswax | 7.70 | (1.45–40.93) | 0.017* |
| Year of introduction in the hive | 2013 | Reference | - | - |
| | 2014 | 2.71 | (0.10–74.55) | 0.55 |
| | 2015 | 2.71 | (0.14–51.60) | 0.51 |
| | 2016 | 2.48 | (0.09–68.14) | 0.60 |
| Location (province) | Antwerp | Reference | - | - |
| | Flemish Brabant | 3.57 | (0.15–85.68) | 0.43 |
| | Walloon Brabant | 0.65 | (0.01–36.56) | 0.84 |
| | Western Flanders | 5.77 | (0.23–143.37) | 0.29 |
| | Eastern Flanders | 6.43 | (0.21–201.07) | 0.29 |
| | Hainaut | 1.67 | (0.06–46.23) | 0.76 |
| | Liège | 3.00 | (0.10–86.09) | 0.52 |
| | Limburg | 2.14 | (0.08–60.17) | 0.65 |
| | Luxembourg | 2.37 | (0.08–66.88) | 0.61 |
| | Namur | 0.56 | (0.01–30.95) | 0.77 |
| Mortality rate (colony level) | Continuous variable | 0.12 | (0.002–9.68) | 0.35 |

*$p$-value less than 0.05.

in Spain by Serra Bonvehí, and Orantes Bermejo [23]. However, in Spain, paraffin adulteration was mostly observed, while in Belgium, stearin/stearic acid adulteration appears to be predominant. This observation is confirmed by the study of Svečnjak et al. [28] which indicates the presence of stearin/stearic acid as adulterant only in Belgium, and The Netherlands amongst the 15 European countries tested. Despite the absence of evidence of a possible effect of the location on the adulteration of beeswax samples (both with paraffin and stearin/stearic acid), if we compare the location of the two different adulterants separately (**Fig 1**), stearin/stearic acid adulteration was exclusively observed in the northern part of the country, whereas, paraffin adulteration was restricted to the southern part. These observations should be in favour of different business networks of adulterated beeswax that need to be further investigated by *ad hoc* authorities to detect the fraud source.

When detecting beeswax adulteration, FTIR-ATR spectroscopy technique has the advantage to detect adulteration at a relatively low level ($< 3\%$) for paraffin, beef tallow, stearin, stearic acid, palmitin, and carnauba wax [39], and its ability to detect mixtures of beeswax adulterants with the same accuracy as single substances [40].

Despite the limited number of beeswax samples from trade (commercial beeswax), 3 out of 9 samples were adulterated by stearin/stearic acid (33.3%). Two of them with a low level ($\leq 3\%$) but one with a high level (20.8%). This last trade beeswax sample was imported from China in 2015. In addition, mosaic brood was reported by several Belgian beekeepers who used wax from this batch when renewing hive foundations. Considering the results of a previous work [30], it is expected that the level of adulteration observed in this survey, could possibly reduce the brood survival rate to less than 55%, confirming the detrimental effect of beeswax adulteration by stearin on bee health.

Beeswax adulteration is an emerging issue and could be a challenge for bee health, as recently shown for stearin, and palmitin [30,41] and possibly for human health too, due to the potential presence of hazardous substances in unrefined paraffin of fossil origin that could be used as adulterant. Carcinogenic compounds such as polyaromatic hydrocarbons (PAHs) are known to be present at substantial concentrations (up to 1%) in unrefined waxes originating from various crude oils [42]. Consequently, the European Commission requested EFSA to define purity criteria for beeswax, and to assess risks for honey bees, and humans [31].

## Conclusion

Beeswax adulteration is a fraud and an emerging issue. It brings the beekeeping sector into disrepute. This survey shows that adulteration by paraffin or stearin/stearic acid in crude beeswax and comb foundation is more frequent than in comb wax. The level of stearin/stearic acid adulterant found is compatible with a detrimental effect on brood. The use of paraffins of petrogenic origin as adulterant must be considered of possible concern for human health, especially for unrefined paraffins that may contain carcinogenic substances such as PAHs, nevertheless, this needs to be properly assessed in the future. There is an urgent need for routine analytical testing of beeswax adulterants and their possible contaminants used in apiculture, in order to produce a regulatory framework that defines beeswax purity criteria, to prevent beeswax adulteration and to ensure the safety of crude, and trade beeswax.

## Supporting information

**S1 Fig. FTIR-ATR spectra of stearic acid *versus* 'stearin' (commercially available as "stearin for candles", a mixture of stearic and palmitic acid) showing the same spectral features.** Wavenumber, the number of waves per unit distance; cm, centimetre; a.u. is for the

absorbance unit.
(TIF)

**S2 Fig.** Comparative spectral features of: An average spectrum of non-adulterated beeswax samples (N = 88) *versus* genuine beeswax (reference standard) [A] an average spectrum of non-adulterated comb foundation samples (N = 6) *versus* genuine beeswax (reference standard) [B]. Wavenumber, the number of waves per unit distance; cm, centimetre; a.u. is for the absorbance unit.
(TIF)

## Acknowledgments

This work would not have been accomplished without the help of the Belgian beekeepers and beekeeping unions that we thank for their time, trust, and for providing us beeswax samples.

## Author Contributions

**Conceptualization:** Noëmie El Agrebi, Agnes Rortais, Jean-Pierre Cravedi, Claude Saegerman.

**Data curation:** Noëmie El Agrebi, Claude Saegerman.

**Formal analysis:** Noëmie El Agrebi, Lidija Svečnjak, Jelena Horvatinec, Véronique Renault, Claude Saegerman.

**Funding acquisition:** Claude Saegerman.

**Investigation:** Noëmie El Agrebi, Lidija Svečnjak, Claude Saegerman.

**Methodology:** Noëmie El Agrebi, Lidija Svečnjak, Agnes Rortais, Jean-Pierre Cravedi, Claude Saegerman.

**Project administration:** Claude Saegerman.

**Resources:** Claude Saegerman.

**Software:** Lidija Svečnjak, Véronique Renault, Claude Saegerman.

**Supervision:** Claude Saegerman.

**Validation:** Lidija Svečnjak, Claude Saegerman.

**Visualization:** Véronique Renault.

**Writing – original draft:** Noëmie El Agrebi, Lidija Svečnjak, Claude Saegerman.

**Writing – review & editing:** Noëmie El Agrebi, Lidija Svečnjak, Jelena Horvatinec, Véronique Renault, Agnes Rortais, Jean-Pierre Cravedi, Claude Saegerman.

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
