## [Decision Letter · Decision Letter 0]

8 Mar 2021

PONE-D-20-37726

Adulteration of beeswax: a first nationwide survey from Belgium

PLOS ONE

Dear Dr. Saegerman,

Thank you for submitting your manuscript to PLOS ONE. After careful consideration, we feel that it has merit but does not fully meet PLOS ONE’s publication criteria as it currently stands. Therefore, we invite you to submit a revised version of the manuscript that addresses the points raised during the review process.

We look forward to receiving your revised manuscript.

Kind regards,

Nicolas Chaline

Academic Editor

PLOS ONE

Journal Requirements:

"The sampling was supported by the Belgian Federal Public Service of Health, Food Chain

355 Safety, and Environment as part of the RF 15/6300 Bee Best Check research project as well as

356 by the SPW ARNE (Service public de Wallonie, Agriculture, Ressources naturelles et

357 Environnement) (Grant RWD32-0286, Bee Tox Check), and the University of Liège (Grant

358 FSR-F-VT-16/16, Bee Tox Check). This work would not have been accomplished without the

359 help of the Belgian beekeepers and beekeeping unions that we thank for their time, trust, and

13

for providing us beeswax samples. The authors wish 360 to thank the University of Zagreb, and the

361 University of Liège for funding the beeswax analyses (both analytical detection and statistical

362 analysis). The authors declare that they have no conflict of interest."

3. We note that Figure 1 in your submission contain map images which may be copyrighted. All PLOS content is published under the Creative Commons Attribution License (CC BY 4.0), which means that the manuscript, images, and Supporting Information files will be freely available online, and any third party is permitted to access, download, copy, distribute, and use these materials in any way, even commercially, with proper attribution. For these reasons, we cannot publish previously copyrighted maps or satellite images created using proprietary data, such as Google software (Google Maps, Street View, and Earth). For more information, see our copyright guidelines: http://journals.plos.org/plosone/s/licenses-and-copyright.

(1) You may seek permission from the original copyright holder of Figure 1 to publish the content specifically under the CC BY 4.0 license. 

Reviewers' comments:

Reviewer's Responses to Questions

**Comments to the Author**

1. Is the manuscript technically sound, and do the data support the conclusions?

Reviewer #1: Yes

2. Has the statistical analysis been performed appropriately and rigorously? 

Reviewer #1: I Don't Know

3. Have the authors made all data underlying the findings in their manuscript fully available?

Reviewer #1: No

4. Is the manuscript presented in an intelligible fashion and written in standard English?

Reviewer #1: Yes

5. Review Comments to the Author

Reviewer #1: The work of Wim Reybroeck was cited under reference [30]. But this is only an abstract of a presentation at the EurBee conference. I think you better refer to the final (ILVO) report:

Reybroeck Wim, Field trial: effect of the addition of a mixture... (June 30, 2017)

https://www.health.belgium.be/sites/default/files/uploads/fields/fpshealth_theme_file/verslag_veldproef_bijenwas_ilvo_rapport_stearine_def_en.pdf

Also the recent publication of Alkassab et al. (2020) in Apidologie should be cited.

file:///C:/Users/ddegraaf/Downloads/Effect_of_contamination_and_adulteration_of_wax_fo.pdf

6. PLOS authors have the option to publish the peer review history of their article (what does this mean?). If published, this will include your full peer review and any attached files.

Reviewer #1: No

---

## [Author Response · Author response to Decision Letter 0]

18 Apr 2021

NOTE OF REVISION

Manuscript: PONE-D-20-37726

Title: Adulteration of beeswax: a first nationwide survey from Belgium

Authors: Noëmie El Agrebi, Lidija Svečnjak, Jelena Horvatinec, Véronique Renault, Agnes Rortais, Jean-Pierre Cravedi, Claude Saegerman

We appreciate the time and efforts by the editor and referees in reviewing this manuscript. We have addressed all issues indicated in the review report. We believe that revised version meet the publication requirements. We have responded specifically to each comment or suggestion below. To make the changes easier to identify where necessary, reviewer’s comments have been numbered. The changes were made within the revised manuscript and were underlined in yellow. A clean version of the manuscript was also added.

EDITORIAL COMMENTS

Comment #1: Please ensure that your manuscript meets PLOS ONE's style requirements, including those for file naming. The PLOS ONE style templates can be found at: https://journals.plos.org/plosone/s/file?id=wjVg/PLOSOne_formatting_sample_main_body.pdf and https://journals.plos.org/plosone/s/file?id=ba62/PLOSOne_formatting_sample_title_authors_affiliations.pdf

Response #1: The PLOS ONE’s style requirements were followed as requested.

Comment #2: Thank you for stating the following in the Acknowledgments Section of your manuscript: “"The sampling was supported by the Belgian Federal Public Service of Health, Food Chain Safety, and Environment as part of the RF 15/6300 Bee Best Check research project as well as by the SPW ARNE (Service public de Wallonie, Agriculture, Ressources naturelles et Environnement) (Grant RWD32-0286, Bee Tox Check), and the University of Liège (Grant 358 FSR-F-VT-16/16, Bee Tox Check). This work would not have been accomplished without the help of the Belgian beekeepers and beekeeping unions that we thank for their time, trust, and for providing us beeswax samples. The authors wish to thank the University of Zagreb, and the University of Liège for funding the beeswax analyses (both analytical detection and statistical analysis). The authors declare that they have no conflict of interest." We note that you have provided funding information that is not currently declared in your Funding Statement. However, funding information should not appear in the Acknowledgments section or other areas of your manuscript. We will only publish funding information present in the Funding Statement section of the online submission form.

Please remove any funding-related text from the manuscript and let us know how you would like to update your Funding Statement. Currently, your Funding Statement reads as follows: "The funders had no role in study design, data collection and analysis, decision to publish, or preparation of the manuscript." Please include your amended statements within your cover letter; we will change the online submission form on your behalf.

Response #2: Funding-related text was removed of the Acknowledgments section. The funders that need to be integrated in the Funding statement was added in the cover letter as requested. Thank you to the Editorial officer in advance if requested changes can be performed by you in the system.

Comment #3: We note that Figure 1 in your submission contain map images which may be copyrighted. All PLOS content is published under the Creative Commons Attribution License (CC BY 4.0), which means that the manuscript, images, and Supporting Information files will be freely available online, and any third party is permitted to access, download, copy, distribute, and use these materials in any way, even commercially, with proper attribution. For these reasons, we cannot publish previously copyrighted maps or satellite images created using proprietary data, such as Google software (Google Maps, Street View, and Earth). For more information, see our copyright guidelines: http://journals.plos.org/plosone/s/licenses-and-copyright. We require you to either

 (1) present written permission from the copyright holder to publish these figures specifically under the CC BY 4.0 license, or (2) remove the figures from your submission: (1) You may seek permission from the original copyright holder of Figure 1 to publish the content specifically under the CC BY 4.0 license. We recommend that you contact the original copyright holder with the Content Permission Form (http://journals.plos.org/plosone/s/file?id=7c09/content-permission-form.pdf) and the following text: “I request permission for the open-access journal PLOS ONE to publish XXX under the Creative Commons Attribution License (CCAL) CC BY 4.0 (http://creativecommons.org/licenses/by/4.0/). Please be aware that this license allows unrestricted use and distribution, even commercially, by third parties. Please reply and provide explicit written permission to publish XXX under a CC BY license and complete the attached form.” Please upload the completed Content Permission Form or other proof of granted permissions as an "Other" file with your submission. In the figure caption of the copyrighted figure, please include the following text: “Reprinted from [ref] under a CC BY license, with permission from [name of publisher], original copyright [original copyright year].”

Response #3: No copyright is needed because: The map was produced by one of the co-authors with quantum-GIS. The GPS data for the country and regional boundaries originate from a copyright free website: DIVA-GIS | free, simple & effective (diva-gis.org). The coordinates of the sample points were collected during the survey and registered into an Excel file. They have been projected with quantum GIS on the country layer and the map. This is therefore an original map with no copyright issues.

Indeed, a sub-section of mapping was added in the M&M section in order to precise this.

REVIEWER #1

GENERAL COMMENT

Comment #1: The manuscript must describe a technically sound piece of scientific research with data that supports the conclusions. Experiments must have been conducted rigorously, with appropriate controls, replication, and sample sizes. The conclusions must be drawn appropriately based on the data presented. 

Respsonse #1: Thank you for your appraisal.

Comment #2: I Don't Know if the statistical analysis been performed appropriately and rigorously.

Response #2: Statistical analysis are appropriately used. This analysis was reviewed by Saegerman C, which is a Professor in Quantitative epidemiology and risk analysis and have large experience and expertise in this matter.

Comment #3: The authors made not all data underlying the findings in their manuscript fully available.

Response #3: In fact, only the precision of the individual results for the level of beeswax adulteration with stearin/stearic acid (N = 7) was not presented in detail (only a range). In the revised version the range was replaced by the individual values as requested. Now, all individual data are presented in the text of the manuscript for completion (i.e. 1.2, 2.2, 2.3, 2.4, 7, 8.1 and 11.9%, respectively).

Comment #4: The manuscript is presented in an intelligible fashion and written in standard English. 

Response #4: Thank you for your appraisal.

SPECIFIC COMMENT

Comment #1: The work of Wim Reybroeck was cited under reference [30]. But this is only an abstract of a presentation at the EurBee conference. I think you better refer to the final (ILVO) report: Reybroeck Wim, Field trial: effect of the addition of a mixture... (June 30, 2017) https://www.health.belgium.be/sites/default/files/uploads/fields/fpshealth_theme_file/verslag_veldproef_bijenwas_ilvo_rapport_stearine_def_en.pdf

Also the recent publication of Alkassab et al. (2020) in Apidologie should be cited.

file:///C:/Users/ddegraaf/Downloads/Effect_of_contamination_and_adulteration_of_wax_fo.pdf

Response #1: The reference of Wim Reybroeck was updated as requested and the recent publication of Alkassab et al. (2020) was added also. Alkassab et al. (2020) becomes the reference [41]. Indeed the previous reference [41] becomes the reference [42] in the revised version.

---

## [Decision Letter · Decision Letter 1]

24 May 2021

Adulteration of beeswax: a first nationwide survey from Belgium

PONE-D-20-37726R1

Dear Dr. Saegerman,

We’re pleased to inform you that your manuscript has been judged scientifically suitable for publication and will be formally accepted for publication once it meets all outstanding technical requirements.

Kind regards,

Nicolas Chaline

Academic Editor

PLOS ONE

Additional Editor Comments (optional):

Reviewers' comments:

Reviewer's Responses to Questions

**Comments to the Author**

1. If the authors have adequately addressed your comments raised in a previous round of review and you feel that this manuscript is now acceptable for publication, you may indicate that here to bypass the “Comments to the Author” section, enter your conflict of interest statement in the “Confidential to Editor” section, and submit your "Accept" recommendation.

Reviewer #1: All comments have been addressed

2. Is the manuscript technically sound, and do the data support the conclusions?

Reviewer #1: Yes

3. Has the statistical analysis been performed appropriately and rigorously? 

Reviewer #1: I Don't Know

4. Have the authors made all data underlying the findings in their manuscript fully available?

Reviewer #1: Yes

5. Is the manuscript presented in an intelligible fashion and written in standard English?

Reviewer #1: Yes

6. Review Comments to the Author

Reviewer #1: The authors have sufficiently responded to the questions raised. I have no further questions or comments.

7. PLOS authors have the option to publish the peer review history of their article (what does this mean?). If published, this will include your full peer review and any attached files.

Reviewer #1: No

---

## [Editor Report · Acceptance letter]

24 Aug 2021

PONE-D-20-37726R1 

Adulteration of beeswax: a first nationwide survey from Belgium 

Dear Dr. Saegerman:

I'm pleased to inform you that your manuscript has been deemed suitable for publication in PLOS ONE. Congratulations! Your manuscript is now with our production department. 

Kind regards, 

on behalf of

Professor Nicolas Chaline 

Academic Editor

PLOS ONE